# Near-Infrared Spectroscopy Procedure for Online Determination of Sodium and Potassium Content in Low-Salt Cured Hams

**DOI:** 10.3390/foods12213998

**Published:** 2023-11-01

**Authors:** María Isabel Campos, Luis Debán, Rafael Pardo

**Affiliations:** 1CARTIF Technology Centre, Agrifood and Sustainable Processes Division, Parque Tecnológico de Boecillo, parcela 205, 47151 Valladolid, Spain; 2Analytical Chemistry Department, Faculty of Sciences, University of Valladolid, Pº de Belén, 7, 47011 Valladolid, Spain; luismariano.deban@uva.es (L.D.); rafael.pardo@uva.es (R.P.)

**Keywords:** dry-cured ham, food sample matrix, NIR spectroscopy, potassium content, sodium content

## Abstract

This paper reports the development of a near-infrared spectroscopy (NIRS) calibration procedure for the determination of sodium and potassium content in cured ham samples. Sliced samples of hams treated with different salts in different percentages were included in the study. Calibration models developed using partial least squares regression were cross-validated and predictive models were tested using the samples of cured ham with low sodium content. The results showed that the developed NIRS procedure is capable of directly measuring the potassium content of packaged dry-cured ham slices with low sodium content with a fitting accuracy of 91.44%, and that it can indirectly determine the sodium content by applying a correction factor to the values obtained for potassium. The prediction error between the calculated and actual sodium values determined using inductively coupled plasma atomic emission spectrophotometry (ICP-AES) was 0.004%, and this confirms that the NIRS procedure is a viable option for the determination of sodium and potassium content in this type of sample.

## 1. Introduction

Methods for the selection of processed meat products based on the rapid detection of critical nutritional parameters are in great demand in the meat industry [1,2]. In particular, the cured ham industry currently requires the use of non-invasive, efficient and objective technologies for evaluating the quality of ham [3,4]. NIR spectroscopy has demonstrated the potential to be used for the real-time automated routine monitorization of dry-cured ham surfaces [5]. In most cases, the purpose of NIR calibration is to replace conventional analytical determination with a procedure that avoids contact with the sample, minimizes the measurement time and enables the quantification of the content of the parameter in each sample of the product in the processing line. Furthermore, this enables the economic break-even point to be reached in mid-term, overcoming the economic impact of conventional analyses [5], and there are environmental advantages, as was reported by Casson et al. in 2020 [6].

However, one of the main limitations in the development of any calibration procedure is the sample matrix. NIR predictive models cannot be directly used for different meat products because they show different matrices and even show important variations within constituent ranges [7]. Using samples that have been subjected to the same process and which have the same origin is essential to obtain a reliable predictive model of the parameter of interest. Therefore, it is essential that each sample present as similar a matrix as possible and that they cover the entire range of variation in the parameter under study [8].

The dependence of NIR calibrations on food sample matrices has been studied in different works for more than three decades, from Begley et al. [9] to Prevolnik et al. [10]. Despite this dependence, calibrations of interest can be achieved, especially from an industrial point of view, and NIR methods that avoid the manipulation of the sample by using remote probes in the quantification of parameters may be capable of replacing the methods currently used [11,12,13,14].

The application of NIR methods to cured ham shows the added complexity of the heterogeneity of the samples, since in the sliced product a whole piece of ham is used, and therefore different muscles are present in its composition. In previous studies, different calibrations have been developed, but in most cases a single muscle has been used [15,16,17]. In these cases, predictive models with high correlation coefficients and low prediction errors were obtained. 

Gou et al. [18] (2013), Campos et al. [19] (2017) and Tejerina et al. [20] (2018) developed applications for the determination of critical parameters in the processing of cured ham with direct relevance for the industry. In these studies, samples of different muscles from white pig hams of different origins subjected to different curing processes were used. This variability in terms of the samples used in calibration was reflected in a decrease in the coefficient of determination and, therefore, in the predictive capacity of the model, although it demonstrated greater applicability in the industry and sufficiently good results to be used in substitution of conventional analytical methods. NIRS technology has advantages for carrying out these types of determinations, e.g., it enables fast and precise measurements in the production line [21], although its limitations when dealing with samples that present high heterogeneity should be taken into account. On the other hand, when NIRS technology is applied to samples that present such heterogeneous matrices, they are subject to the wet method performed during the construction of the model in terms of the thickness of the samples [22]. 

The significance of having a technology that enables the monitoring of critical parameters, not only for the control of industrial food processes but also for consumers, must be emphasized. It is well established that excessive sodium intake is directly linked to an increased risk of cardiovascular disease, and thus, it is imperative to limit such intake in order to reduce the incidence of coronary heart disease and strokes [23,24,25].

In July 2012, the Spanish Agency for Food Safety and Nutrition (AESAN), together with the Spanish Confederation of Meat Retailers (CEDECARNE) and the Association of Manufacturers and Marketers of Food Additives and Supplements (AFCA), reached an agreement to reduce salt in butchery–charcuterie products. This agreement is in line with scientific evidence that supports the relationship between excessive sodium consumption and morbidity and mortality from cardiovascular diseases and other chronic diseases. Furthermore, this agreement is part of the recommendations for the reduction of selected nutrients promoted by the European Union and the World Health Organization.

One method that can be used to reduce sodium in the curing of ham is partial substitution with other salts. Specifically, the partial replacement of sodium with potassium is a practice that has been carried out by several researchers in recent years, and this has resulted in sodium reductions of up to 30% and the production of cured hams without microbial or proteolytic alterations and with adequate texture, flavor and smell [26,27,28,29].

The use of near-infrared spectroscopy to monitor sodium and potassium salts in cured ham provides immediate control over the technological characteristics of its production and, likewise, is very positive for the consumer, who as a result can know the sodium content of a product and therefore make informed choices according to their health and preferences (i.e., higher or lower salt content).

In this work, the influence of changes in salting during the processing of cured ham on the development of NIRS calibrations were studied. For this purpose, ham slices of different parts of ham piece that had been subjected to different salting processes were used: (a) hams salted only with sodium chloride and (b) hams with low sodium content treated with mixtures of sodium and potassium salts (NaCl and KCl) in different percentages. It was observed that the samples cured with this mixture were grouped outside the initial calibration set. We were able to accurately determine the potassium content and, by using the corresponding correction factor, determine the sodium content.

The developed model enables the online determination of the sodium and potassium content of a company’s production line and the classification of cured ham containers according to their sodium content.

## 2. Materials and Methods

### 2.1. Samples

This study was carried out on sliced samples of each muscle from a whole piece of cured ham (white pig hams) treated with different salts during the salting proceses. Two groups of samples were selected. Group 1 (G1) consisted of sliced ham samples obtained using the usual curing process in which only sodium was used in the salting process. The second group (G2) consisted of hams cured with sodium and potassium in similar amounts. All the samples were frozen at −12 °C until the NIR measurements were carried out. The measurements, both spectroscopic and analytical, were carried out at CARTIF Technology Centre.

### 2.2. Spectra Acquisition

Spectroscopic measurements were carried out via interactance–reflectance (between 12,000 and 4000 cm^−1^) using a Fourier transform (FT) NIR spectrometer model Matrix-F emission equipped with a non-contact measurement fiber optic probe Q-412/AF (Bruker Optik GmbH, Ettlingen, Germany). 

The spectroscopic measurements of the samples were collected directly from the slices of cured ham by adjusting the height of the tray to keep a constant optical path length of 10 cm. The reflectance spectra were collected from a circular sample area (diameter = 10 mm) in a central zone in the middle of the slice. Two spectra of each sample were obtained at a temperature between −12 and −10 °C using a resolution of 16 cm^−1^, performing 16 scans and spending 5.01 s on each measurement.

### 2.3. Chemical Analysis 

The sodium and potassium content analysis was carried out according to the method outlined by Campos et al. (2017) [19]. The areas irradiated by the NIR probe in the first six half-slices of each tray were minced and homogenized and then digested using an HCl-HNO3 solution (Merck Group, Madrid, Spain) (9:1) at 90 °C for 45 min. The sodium and potassium content was then determined via inductively coupled plasma atomic emission spectrophotometry (ICP-AES) using a 720-ES Varian spectrometer (Varian, Vista, Australia). Calibration curves were determined using a 1000 ppm sodium and potassium standard. 

### 2.4. Multivariate Analysis

For the development of the predictive models, the chemometrics software OPUS/QUANT™ version 7.0 was used. A partial least squares regression (PLSR) analysis and a cross-validation method were applied. Outliers were identified and removed from the model using Mahalanobis distance [9]. 

The prediction abilities of the developed NIR methods were evaluated using the following measurements: standard error obtained through cross-validation (root mean square error of cross-validation, RMSECV); coefficient of determination for cross-validation (R^2^cv) determined using the reference values (real values) against the values obtained via the NIR models (predicted values); predictive capacity (residual prediction deviation, RPD); and standard error obtained through external validation (root mean square error of prediction, RMSEP), calculated in the same way as its homologue in calibration, but using the values from the external validation set.

## 3. Results and Discussion

### 3.1. Chemical Analysis

Table 1 shows the statistics descriptors of both sets of samples. The samples corresponding to G1 contained sodium in a range between 1.21 and 3.10%. The sodium content of the G2 samples was lower than that of the G1 samples, in a range between 0.77 and 1.25%, due to the fact that in the curing of these samples, the sodium content was reduced by partially replacing it with potassium. The potassium content of the G2 samples was between 1.04 and 1.80%.

### 3.2. Spectral Analysis

This study was carried out using the first derivatives of the average spectra of the samples from both groups, the G2 samples salted with sodium and potassium and the G1 samples treated only with sodium. For the purpose of comparison within the G1 group, two samples with very different sodium content were chosen for the study: G1-1 (2.97% Na) and G1-2 (1.21% Na). 

A 1984 study by Begley et al. [9] concluded that changes induced by salt in the water spectrum can be isolated from other spectral variations via mathematical treatment, there being a high correlation between salt content determined using chemical analysis and salt content determined using NIR spectra at a frequency of 5537 cm^−1^. Figure 1 shows that at this frequency the values obtained for the samples cured with sodium and potassium salts, corresponding to the first derivative, are within a range that does not correspond to the sodium concentration acquired using ICP-AES (0.93% Na). 

### 3.3. NIR Calibration Results—Valuation of Predictive Models: Cross-Validation

Calibration was performed via cross-validation using both sets of samples. Using the predictive model for sodium that provided the best results, a correlation coefficient (R^2^cv) of 85.13%, a RMSECV of 0.19% Na and an RPD of 2.59 were obtained.

In general, there was a very good concordance between the values obtained using the reference method (x axis) and the theoretical values supplied by the model (y axis), though this was not the case for the G2 samples (Figure 2a). The line represents a 1:1 relation between the x and y values, and the different G2 sample colors (orange and blue) indicate different Na/K ratios.

When building the model, as is shown in Figure 2a, the G2 samples showed a clear deviation with respect to the calibration set.

### 3.4. External Validation of Samples from Different Matrix Using the Predictive Model

According to the results obtained from the previous calibration that included samples from both groups, an external validation of the G2 samples was carried out via a calibration performed using the samples treated only with sodium belonging to the group G1. The values obtained from this external validation are shown in Table 2. In each sample, the predicted value was higher than the Na value present in the sample, as measured using ICP-AES. However, these values correspond to the K values obtained when analyzing the samples via ICP-AES, and this must be attributed to the fact that the samples had been cured using a mixture of sodium and potassium (Figure 2a).

To confirm the above, a study was performed on the Na and K content used in the process of salting the hams, taking into account the fact that the G2 slices were cured by partially replacing the sodium with potassium. Table 3 shows the values collected from the analysis of the sodium and potassium, both of which were measured using ICP-AES, in addition to the total percentages of both metals and the predicted values after the external validation of the Na.

The sum of the average sodium and potassium values obtained via wet chemistry for the G2 samples was 2.39%, a value that is within the range of the spectra with sodium values between 1.99 and 2.97% (Figure 1).

However, the model predicted a value for these samples of 1.47%, and that corresponded to the percentage of potassium analyzed. This fact can be explained by the greater penetrability of the K+ ions in the muscles compared to the Na+ ions [30]; this implies that in the sample slices the majority of the salt identified via NIRS corresponded to potassium. This was verified by the results shown in Table 2, which were obtained by comparing the sodium and potassium values of each sample provided by the model. If it is assumed that the predicted value refers to Na, the prediction error obtained through external validation (RMSEP) is 0.31%, which is far from what the model allows, i.e., no more than 0.19% of the RMSECV, as indicated in the previous section. On the other hand, if the value predicted by the model is the percentage of potassium, the prediction error decreases considerably to 0.01%.

This enables the direct measurement of potassium content using NIRS and the indirect measurement of sodium content by applying to the NIRS results an estimated correction factor of 0.63, which corresponds to the ratio between the percentages of Na analyzed using ICP-AES and that of K obtained using the NIRS method. In all cases, once the correction was applied, the prediction error between the calculated and actual sodium values (ICP-AES) was 0.004%, which implies the viability of the procedure for the determination of Na and K in this type of sample. This was verified in the NIRS model (Figure 2b), in which a fit of 91.44% R2cv, an RMSECV of 0.13% and an RPD of 3.42 were obtained.

## 4. Conclusions

The NIRS procedure developed herein enables the direct measurement of potassium and the indirect measurement of sodium in ham samples cured with sodium and potassium salts that are marketed as low-sodium cured ham. The predictive model achieved a fit of 91.44% in terms of the coefficient of determination for cross-validation, a predictive capacity of 3.42 and a standard prediction error of 0.13% Na. The prediction error between the calculated and actual sodium values (ICP-AES) was 0.004%, which verifies the feasibility of the procedure for the determination of Na and K in this type of sample. These results suggest that online NIRS technology is a suitable tool which can be implemented on the packaging line for dry-cured ham slices and thus obtain accurate and relevant information on the potassium and sodium content of each packaged product.

## Figures and Tables

**Figure 1 foods-12-03998-f001:**
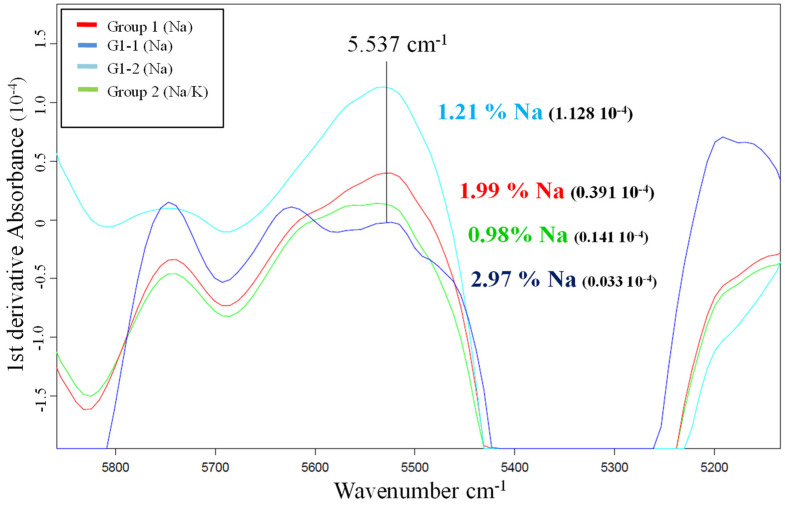
First derivatives of the average spectra of the samples with different sodium percentages.

**Figure 2 foods-12-03998-f002:**
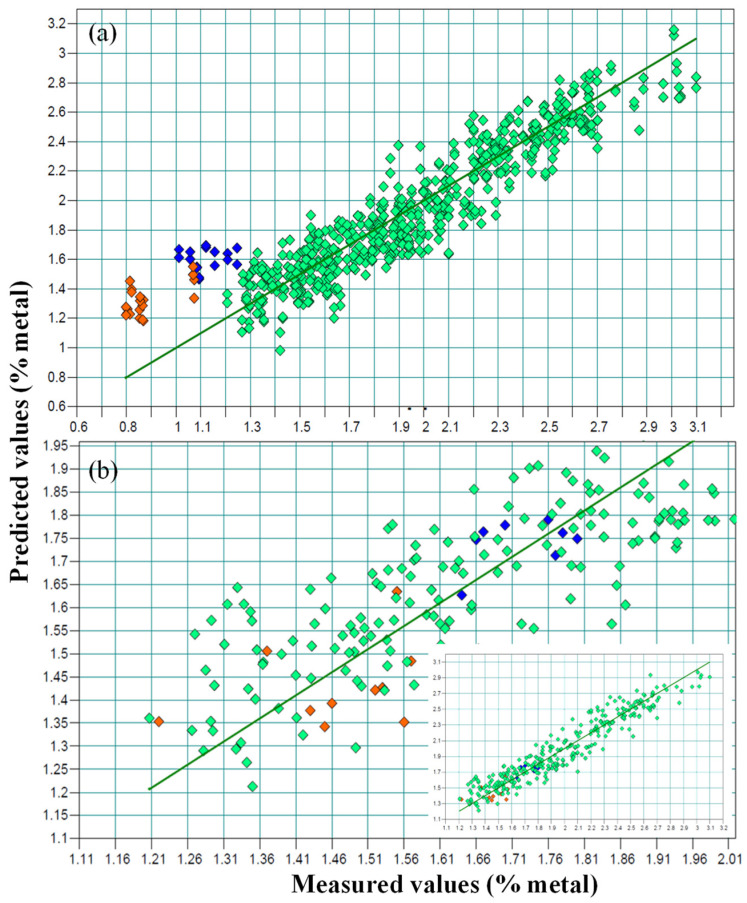
NIR prediction plots: predicted versus real values, including the samples with reduced sodium content (blue and orange colors, represent samples treated with different formulations with reduced sodium content). (**a**) Sodium values as percentages; (**b**) the real values of the G2 samples correspond to the potassium content (%).

**Table 1 foods-12-03998-t001:** Statistical overview of chemical analysis of sodium and potassium content.

Samples	No. of Samples	Sodium				Potassium			
		Mean	Range	SD ^1^	CV ^2^ (%)	Mean	Range	SD ^1^	CV ^2^ (%)
G1 (Na)	310	1.99	1.21–3.10	0.45	22.81	-	-	-	-
G2 (Na/K)	60	0.93	0.77–1.25	0.11	11.86	1.47	1.04–1.80	0.21	14.59

^1^ SD, standard deviation; ^2^ CV, coefficient of variation (SD * 100/mean).

**Table 2 foods-12-03998-t002:** Differences between the real and predicted metal content values of the G2 samples obtained via external validation.

G2 Sample	Real Value			External Validation ^1^		
	Na (%)	K (%)	Na + K (%)	Predicted Value	CF ^2^	Calculated Sodium Value ^3^ (%)
N1	1.21	1.66	2.87	1.76	0.69	1.11
N2	1.06	1.8	2.86	1.75	0.61	1.1
N3	1.09	1.64	2.73	1.63	0.67	1.03
N4	1.12	1.76	2.88	1.8	0.62	1.13
N5	1.07	1.37	2.44	1.53	0.7	0.96
N6	0.82	1.57	2.39	1.49	0.55	0.94
N7	0.85	1.43	2.28	1.37	0.62	0.86
N8	0.8	1.53	2.33	1.36	0.59	0.86
N9	0.81	1.67	2.48	1.44	0.56	0.91
N10	1.25	1.77	3.02	1.79	0.7	1.13
N11	1.16	1.78	2.94	1.7	0.68	1.07
N12	1.08	1.7	2.78	1.74	0.62	1.1
N13	1.01	1.56	2.57	1.78	0.57	1.12
N14	0.87	1.52	2.39	1.35	0.64	0.85
N15	0.86	1.22	2.08	1.41	0.61	0.89
N16	0.8	1.46	2.26	1.37	0.58	0.86
N17	0.85	1.55	2.4	1.43	0.59	0.9
N18	1.07	1.73	2.8	1.63	0.66	1.03
N19	0.87	1.33	2.2	1.36	0.64	0.86
N20	0.91	1.28	2.19	1.34	0.68	0.84
N21	0.88	1.23	2.11	1.28	0.69	0.81
N22	0.83	1.23	2.06	1.31	0.63	0.83
N23	1.04	1.61	2.65	1.63	0.64	1.03
N24	1.14	1.68	2.82	1.73	0.66	1.09
N25	0.99	1.51	2.5	1.58	0.63	1
N26	0.81	1.27	2.08	1.33	0.61	0.84
N27	0.89	1.28	2.17	1.43	0.62	0.9
N28	1.07	1.76	2.83	1.81	0.59	1.14
N29	1	1.7	2.7	1.72	0.58	1.08
N30	1.01	1.46	2.47	1.55	0.65	0.98
N31	1.03	1.58	2.61	1.6	0.64	1.01
N32	0.84	1.39	2.23	1.42	0.59	0.89
N33	0.95	1.59	2.54	1.7	0.56	1.07
N34	0.91	1.4	2.31	1.44	0.63	0.91
N35	0.8	1.21	2.01	1.32	0.61	0.83
N36	0.88	1.24	2.12	1.29	0.68	0.81
N37	0.83	1.15	1.98	1.23	0.67	0.77
N38	0.77	1.18	1.95	1.25	0.62	0.79
N39	0.81	1.12	1.93	1.18	0.69	0.74
N40	0.97	1.58	2.55	1.64	0.59	1.03
N41	1.04	1.51	2.55	1.57	0.66	0.99
N42	1	1.49	2.49	1.54	0.65	0.97
N43	0.92	1.4	2.32	1.45	0.63	0.91
N44	0.87	1.37	2.24	1.43	0.61	0.9
N45	0.92	1.53	2.45	1.56	0.59	0.98
N46	0.87	1.12	1.99	1.21	0.72	0.76
N47	0.92	1.15	2.07	1.17	0.79	0.85
N48	0.87	1.13	2	1.19	0.73	0.75
N49	0.92	1.04	1.96	1.15	0.8	0.84
N50	0.8	1.08	1.88	1.19	0.67	0.75
N51	0.91	1.25	2.16	1.32	0.69	0.83
N52	0.86	1.2	2.06	1.27	0.68	0.8
N53	0.98	1.48	2.46	1.51	0.65	0.95
N54	0.9	1.56	2.46	1.63	0.55	1.03
N55	0.9	1.34	2.24	1.43	0.63	0.9
N56	0.9	1.42	2.32	1.45	0.62	0.91
N57	0.89	1.55	2.44	1.58	0.56	1
N58	1.1	1.66	2.76	1.75	0.63	1.1
N59	1.06	1.75	2.81	1.83	0.58	1.15
N60	0.8	1.26	2.06	1.3	0.62	0.82

^1^ External validation of predictive model developed using the G1 samples as the calibration set; ^2^ CF, correction factor: ratio between real sodium value and value predicted in external validation (%); ^3^ value predicted in external validation * 0.63.

**Table 3 foods-12-03998-t003:** Real and predicted values of metal content of samples cured with different salts.

Samples		No. of Samples	Real Value			Predicted Value (%)
			Na (%)	K (%)	Na + K (%)	
G1 (Na)	Mean	310	1.99	-	-	1.96
	G11	1	2.97	-	-	2.75
	G12	1	1.21	-	-	1.34
G2 (Na/K)	Mean	60	0.93	1.46	2.39	1.47

## Data Availability

The data presented in this study are available on request from the corresponding author. The data are not publicly available due to the confidentiality of the industrial process of the Company.

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
