# Peer review of "Near-Infrared Spectroscopy Procedure for Online Determination of Sodium and Potassium Content in Low-Salt Cured Hams"

_foods, 2023, doi:10.3390/foods12213998_

Round 1

Reviewer 1 Report

Comments and Suggestions for Authors

line 80. Please specify the samples of hams you've used. Did they contain different muscles in one piece, or one muscle?

line 83. "similar amounts" - please explain. Earlier (lines 72-74) you have mentioned that hams were treated with mixtures of sodium and potassium salts  in different percentages. How many mixtures you've used for salting  in this experiment?

lines 117-120. Dear Authors, you have written the same data as were presented in the Table 1. Please describe these results the other way.

Table 1. Why the numbers of sample G1 and G2 are so different?  

Fig 1 and Fig.2 should be improved. They do not show the data properly.

Lines 192 Conclusion. Please add more arguments to prove the method developed is applicable. The Manuscript is entitled "Online estimation for sodium and potassium control in low-salt cured hams". How long it takes to measure one piece? Does the ham piece thickness influence the result? How thick could be a slice? How low can be Na content to be detected in ham? What is sodium content in a cured ham marked as "low-sodium"? Some data you've described in Campos, M. I., Debán, L., Antolín, G., & Pardo, R. (2023). A quantitative on-line analysis of salt in cured ham by near-infrared  spectroscopy and chemometrics. Meat Science, 200, 109167. Doi.10.1016/j.meatsci.2023.109167 What new is in this Manuscript?

Author Response

Dear Reviewer,

We would like to thank you very much for the revision of the submitted manuscript. In order to facilitate the communication, we have listed all the requirements from the reviewers (in italics) and have responded one by one (in green letter).

Kind regards,

Maribel Campos

Reviewer 2 Report

Comments and Suggestions for Authors

The article present a new on-line method of non-contacting determining NaCl and KCl in cured ham samples by NIR spectroscopy technique. This method can be used for the measurements of salt content in each sample of the product and in processing line. In general, in the presence of both salts, the developed calibration technique allows first determine the potassium concentration, and then, taking into account the obtained value, determine the sodium concentration.  The authors use a derivative of the NIR spectra, which probably makes it possible to distinguish sodium and potassium signals to some extent, as well as reduce the matrix contribution.

I think that the article can be improved taking into account the following comments and suggestions:

- The drawings are presented in the manuscript in a low resolution and, of course, they cannot be published in this form. I hope the authors have the drawings in a normal form.

- It is not clear from the Figure 1 in what quantities the first derivative is plotted.

- The procedure for the averaging NIR spectra must be described. This procedure determines the accuracy of determining the first derivative. It is necessary to provide the typical error in determining the derivative of spectra at the wavelength used.

- In Figure 2, the authors should indicate concentrations instead of values.  

Author Response

(The authors gave the same response as above.)

Reviewer 3 Report

Comments and Suggestions for Authors

This article is written on (Online estimation for sodium and potassium control in low salt cured hams), however here are a few suggestions that will improve the quality of the manuscript if followed by the authors

1.      Need improvement in title as its not clear statement.

2.      Needs improvement in abstract, more focused on the aim and methodology and conclusion part of the abstract. Also add numerical values of the results in abstract section.

3.      Line 12 and 13 please add names of salts with their percentages.

4.      Add a conclusive line at the end of the abstract

5.      Mention novelty in this study in the abstract

6.      After calibration and validation of regression model please explain the statistical analysis in one line in the abstract, how you have interpreted the results.

7.      Keywords should be written in alphabetical order

8.      In the introduction, in-cite references should be the latest and focused on intake of sodium and potassium intake.

9.      Please write a paragraph about the rational, importance and aim of the study at the end of introduction section,

10.  Keep the beginning with recent supported findings. If you can find a related intro of the main title published in latest years, that would be preferable.

11.  Please concise your introduction section with sequence, give some back ground information about the intake of low salt food matrix, then explain the effects of NIR and then make a sequenced aim in the end.

12.  Line 78. Material. The authors did not mention where they arranged chemicals for the present study.

13.  Please add a reference in the section 2.2 which method you have followed.

14.  Please mention the sample size in the methodology section.

15.  Line 136: Figure 1 is not clear, even it is not readable, please attach it clearly.

16.  Line 146: Figure 2 is also not clear, even it is not readable, please attach it clearly.

17.  Authors should provide proper justification and reasoning in each parameter along they should compare with previous studies.

18.  Need Improvement in conclusion with mean values and provide in-depth information of this study. Can you give suggestions to other researchers, who are conducting the research in the same domain as yours?

19.  Please mention the full form first and then abbreviations in the following text of the manuscript.

20.  Grammar needs serious attention; a lot of sentences have no sense.

Comments on the Quality of English Language

Extensive editing of English language required

Author Response

(The authors gave the same response as above.)
